# A Boronated Derivative of Temozolomide Showing Enhanced Efficacy in Boron Neutron Capture Therapy of Glioblastoma

**DOI:** 10.3390/cells11071173

**Published:** 2022-03-31

**Authors:** Jing Xiang, Lin Ma, Zheng Gu, Hongjun Jin, Hongbin Zhai, Jianfei Tong, Tianjiao Liang, Juan Li, Qiushi Ren, Qi Liu

**Affiliations:** 1Department of Biomedical Engineering, College of Engineering, Peking University, Beijing 100871, China; jenny-94@163.com; 2Shenzhen Bay Laboratory, Institute of Biomedical Engineering, Shenzhen 518132, China; guzheng@szbl.ac.cn; 3Department of Stomatology, General Hospital, Shenzhen University, Shenzhen 518055, China; malin2021@szu.edu.cn; 4Guangdong Provincial Key Lab of Biomedical Imaging, The Fifth Affiliated Hospital of Sun Yat-sen University, Zhuhai 519000, China; jinhj3@mail.sysu.edu.cn; 5Institute of Biomedical Engineering, Peking University Shenzhen Graduate School, Shenzhen 518055, China; zhaihb@pku.edu.cn; 6Institute of High Energy Physics, Chinese Academy of Sciences (CAS), Beijing 100049, China; tongjf@ihep.ac.cn (J.T.); liangtj@ihep.ac.cn (T.L.); lijuan@ihep.ac.cn (J.L.); 7Spallation Neutron Source Science Center, Dongguan 523803, China

**Keywords:** boron neutron capture therapy, glioblastoma, boron delivery agents, binary radiation therapy

## Abstract

There is an incontestable need for improved treatment modality for glioblastoma due to its extraordinary resistance to traditional chemoradiation therapy. Boron neutron capture therapy (BNCT) may play a role in the future. We designed and synthesized a ^10^B-boronated derivative of temozolomide, TMZB. BNCT was carried out with a total neutron radiation fluence of 2.4 ± 0.3 × 10^11^ n/cm^2^. The effects of TMZB in BNCT were measured with a clonogenic cell survival assay in vitro and PET/CT imaging in vivo. Then, ^10^B-boronated phenylalanine (BPA) was tested in parallel with TMZB for comparison. The IC50 of TMZB for the cytotoxicity of clonogenic cells in HS683 was 0.208 mM, which is comparable to the IC50 of temozolomide at 0.213 mM. In BNCT treatment, 0.243 mM TMZB caused 91.2% ± 6.4% of clonogenic cell death, while 0.239 mM BPA eliminated 63.7% ± 6.3% of clonogenic cells. TMZB had a tumor-to-normal brain ratio of 2.9 ± 1.1 and a tumor-to-blood ratio of 3.8 ± 0.2 in a mouse glioblastoma model. BNCT with TMZB in this model caused 58.2% tumor shrinkage at 31 days after neutron irradiation, while the number for BPA was 35.2%. Therefore, by combining the effects of chemotherapy from temozolomide and radiotherapy with heavy charged particles from BNCT, TMZB-based BNCT exhibited promising potential for therapeutic applications in glioblastoma treatment.

## 1. Introduction

Glioblastoma (GBM) is aggressive, resistant to therapy, easy to relapse, and associated with a very poor prognosis [1]. Indeed, little progress has been made in the last century in terms of GBM prognosis. A study of GBM patients suggested a median survival of 8.3 months in 1950 [2]. By contrast, GBM remains incurable with a median survival of 15 months decades later [3,4]. Thus, there is an incontestable need for improved treatment modality for GBM patients.

A consensus guideline for the standard of care (SOC) for GBM has been made based on extensive clinical results [5,6,7]. To decrease the infiltration of the eloquent regions in the brain of GBM patients, a maximum safety surgical resection of bulk disease is performed. Subsequently, involved-field radiotherapy with 60 Gy x-rays delivered over six weeks in 30 fractions with concurrent daily temozolomide (TMZ) is given, followed by a few months of adjuvant TMZ [3,8]. Because GBM is a diffuse tumor type and surgery is not able to completely remove it, radiotherapy is used to capture the unresected infiltrating GBM cells by encompassing at least 2 cm in all dimensions beyond tumor margins [9]. However, approximately 90% of GBM still recurs in the region close to the original tumor site [10,11]. Therefore, novel treatment strategies are developed to overcome the extraordinary therapeutic resistance of GBM [9,12,13,14,15,16,17,18]. Notably, binary radiation therapy using small molecules, biologics, or nanomedicine to improve radiation effects in GBM is increasingly tested in clinical or preclinical studies [13,14,15,16,17,18], where boron neutron capture therapy (BNCT) has attracted a lot of attention [19,20].

BNCT combines the advantages of cancer-targeting drugs and heavy ion radiation therapy for the effective elimination of cancer cells [19,20]. BNCT requires boron-10 (^10^B)-enriched drugs that can transport ^10^B isotopes and accumulate in tumor cells selectively. Moreover, ^10^B has a very large thermal neutron cross-section (3830 barn) and can efficiently capture thermal neutrons (1 eV–10 keV), leading to a nuclear reaction [21]:B 105+n 1o(th)→[B 115]*→H 4e2(α)+L 7i+2.38 MeV

When cancer cells containing a sufficient concentration of ^10^B are irradiated with thermal neutrons, the nuclear reaction will occur and release α and ^7^Li^3+^ particles. These heavy charged particle rays have very high LET values (~230 keV/μm) and short track lengths (about 10 μm), which can generate densely ionizing events along their tracks in a single cell scale, leading to complex DNA damages and efficient cell killing [22,23]. Compared with traditional radiotherapy, BNCT allows the preferential destruction of ^10^B-containing cells, while thermal neutrons alone, without nuclear reactions with ^10^B, cause little damage. Thus, with tumor-targeting boron delivery agents, BNCT may generate more irreparable damage to cancer cells, but induce little damage to surrounding normal tissues, thereby supposedly leading to increased tumor control probability.

The successful application of BNCT depends on boron delivery agents that can reach approximately 20 µg ^10^B per gram of tumor and tumor-to-normal tissue ratios at at least 3:1 during neutron irradiation [24]. The development of a boronated compound that has a high uptake in cancer cells while avoiding significant uptake and toxicity in normal tissues is quite challenging [19]. The commonly used boron delivery agents in BNCT are boronated phenylalanine (BPA) and sodium borocaptate (BSH) [25]. Because BSH cannot penetrate the blood–brain barrier, BPA has been the most used drug in testing the BNCT treatment of GBM [26]. Meanwhile, more and more studies are exploring novel boron delivery agents with better performance in GBM treatments [19,27].

TMZ is the first-line chemotherapy drug for GBM treatment [28]. As a small lipophilic molecule in the imidazotetrazine class, TMZ can penetrate the blood–brain barrier and induce DNA lesions by alkylating the purine bases of DNA [29]. To exploit the advantages of TMZ for BNCT, we constructed a ^10^B-labeled derivative of TMZ, named TMZB, in this study. We expected that TMZB can carry ^10^B atoms through the blood–brain barrier and reach GBM cells, while the synthesis method does not affect the splitting of the diazo group on the left side of the compound during TMZ activation, and so the cytotoxic effect of TMZ will be retained during BNCT. We tested a variety of synthetic routes and products with different chemical structures, and identified the following compound to serve our purpose, i.e., 4-dihydro-3-methyl-4-oxoimidazo[5,1-D]-1,2,3,5-tetrazine-8-carboxylic acid [4-(4,4,5,5-tetramethyl-1,3,2-diazaboran-2-yl) phenyl ester. We measured the in vitro and in vivo effects of TMZB treatment alone or in combinational use with neutron irradiation in BNCT with GBM models. TMZB performed better than BPA in most tests. Our work provides a dual-function agent for ^10^B delivery and GBM toxicity at the same time. The improved efficacy of BNCT observed in this study guarantees further investigation in clinical settings for GBM treatment with TMZB.

## 2. Materials and Methods

### 2.1. Reagents and Equipment

All reagents and solvents were purchased from commercial sources and used following validated protocols. High-performance liquid chromatography (HPLC) was performed using Shimadzu’s LC-40 (Shimadzu Corporation, Koyoto, Japan). Boron concentrations were analyzed using an ICP-AES (Teledyne Leeman Labs, Hudson, NH, USA). In addition, ^1^H NMR spectra were recorded on a WNMR-I-400MHz (Wuhan Zhongke Co., Ltd., Wuhan, China). Moreover, ^13^C NMR was recorded on a Bruker-600MHz and this was calibrated using a residual undeuterated solvent as an internal reference (CDCl_3_: ^1^H NMR *δ* = 7.26 ppm, ^13^C NMR *δ* = 77.16 ppm). Neutron radiation was generated from a radio frequency quadrupole (RFQ) accelerator with a high-current proton beam at the Spallation Neutron Source Science Center, Dongguan, China. The proton beam from the accelerator passes through the beam transmission line and emits lithium. The final beam output power is 35 kW, the beam energy is 3.5 MeV, the pulse current intensity is 30 mA, the average current intensity is 10 mA, the beam duty cycle is 33.3%, and the center frequency of the beam is 352.2 MHz.

### 2.2. Cell Culture

Human GBM cells U87MG, U251, and HS683 and endothelial cell HUV-EC were obtained from the National Infrastructure of Cell Line Resource in Beijing, China. Cell lines were authenticated at the National Infrastructure of Cell Line Resource, Beijing, China. GBM cells were cultured in a DMEM medium containing 10% fetal bovine serum and 1% penicillin/streptomycin at 5% CO_2_ and 37 °C. HUV-EC cells were cultured in complete culture medium (Procell Life Science & Technology Co., Ltd., Wuhan, China). Cells were tested mycoplasma free using a Mycoplasma Test Kit (Cellorlab, China). All materials for cell culture were purchased from Gibco Invitrogen (Grand Island, NY, USA) unless otherwise noted.

### 2.3. GBM Mouse Model

Single-cell suspension with HS683 cells was prepared at 2 × 10^7^ cells/mL. Nude mice were anesthetized with an intraperitoneal injection of tribromoethanol at a dose of 150 mg/kg and positioned in a stereotaxic instrument. After disinfection with 75% alcohol, the scalp was cut open with a scalpel, the bregma point was selected, and a hole was made in the mouse skull using a 1 mm cranial drill. Then, 5 μL of cell suspension was taken using a micro-injector, which was then fixed on the brain with a needle depth at 3.0 mm using a stereotaxic instrument. Three minutes later, the micro-injector was lifted by 0.5 mm to make rooms for the injection of cells at 1 μL/min. After removing the needle, the drill hole in the mice skull was then filled with bone wax, the skin incision was sutured, and the wound was disinfected with iodophor. The GBM model was established approximately 28 days after cell injection.

### 2.4. Syntheses of TMZB

As shown in Figure 1, to a 50 mL round-bottom flask loaded with 3,4-dihydro-3-methyl-4-oximidazo [5,1-D]-1,2,3,5-tetrazine-8-carboxamide acid (200 mg, 1.02 mmol), 4-bromomethylboronic acid pinacol ester (600 mg, 2.02 mmol) and Na_2_CO_3_ (170 mg, 1.60 mmol), DMF (5 mL) were added. The mixture was stirred at 55 degrees for 60 min. The reaction mixture was monitored using TLC and was subsequently cooled to room temperature. After dilution with water (30 mL) and saturated salt water (30 mL), the mixture was partitioned using ethyl acetate. The solvent was removed under vacuum in a rotatory evaporator and the residue was dissolved in DCM/MeOH, and then purified by column chromatography to obtain a white solid of compound **B** (40 mg, yield 10%). (^1^H NMR (600 MHz, chloroform-d) *δ* 8.46 (s, 1H), 7.84–7.81 (m, 2H), 7.53 (d, J = 7.9 Hz, 2H), 5.54 (s, 2H), 4.05 (s, 3H), 1.34 (s, 12H)).

Compound **A** mixture containing compound **B** (200 mg, 0.486 mmol), methylboronic acid (300 mg, 5 mmol), and trifluoroacetic acid in a solvent (5% solution) of dichloromethane (5 mL) was stirred overnight at room temperature. The solvent was removed under vacuum in a rotatory evaporator and the residue was dissolved in DCM/MeOH and the crude product was purified with column chromatography to afford the product **C** (130 mg, yield 65.0%) as a white solid. (^1^H NMR (600 MHz, chloroform-d) *δ* 8.46 (s, 1H), 7.84–7.81 (m, 2H), 7.53 (d, J = 7.9 Hz, 2H), 5.54 (s, 2H), 4.05 (s, 3H), 1.34 (s, 12H)).

To a solution of **C** (100 mg, 0.304 mmol) in THF (3 mL) was added dropwise with methyl diethanolamine (36.2 mg, 0.304 mmol) and stirred at room temperature for 4 h. The mixture was concentrated under reduced pressure and a low temperature. The crude product was dissolved with deuterated chloroform and the solvent was removed under vacuum in a rotatory evaporator to afford the product, TMZB (compound **D**) as a white solid (100 mg, yield 79.8%). (^1^H NMR (400 MHz, chloroform-d) *δ* 8.85 (s, 1H), 7.54–7.48(m, 2H), 7.34 (d, J = 7.8 Hz, 2H), 5.39 (s, 2H), 3.97–3.84 (m, 7H), 3.24 (ddd, J = 11.6, 5.4, 4.1 Hz, 2H), 2.95 (ddd, J = 11.6, 8.3, 6.5 Hz, 2H), 2.20 (s, 3H); HRMS [30] C_24_H_21_BN_6_O_5_[M + H]+: 415.17, found: 415.21. Decomposition at about 200 °C).

### 2.5. Boron Concentration Analyses

TMZB (molecular weight, 412.21) and BPA (209.01; obtained from HEC Pharm, Dongguan, China) were made into 0.0–1.0 mg/g aqueous solutions. GBM cells and HUV-EC cells were treated with different concentrations of TMZB or BPA solutions. The drug-containing medium was removed 12 h later. The biodistribution of boron in TMZB- or BPA-treated mice was measured at the timepoint for neutron irradiation, i.e., 2 h after drug injection [31]. Eight nude mice in each group were injected with 500 mg/kg of BPA, TMZ, or TMZB through the tail vein of the nude mice, in a 3,5-tetrazine-8-methoxy-4-phenylboronic acid-ethyl ester solution. Tissues of the nude mice (heart, liver, spleen, lung, kidney, and tumor site) were subsequently collected within 2 h. Cells or tissues were digested with concentrated nitric acid for the analysis of boron concentration via ICP-AES.

### 2.6. Cell Proliferation Assay

Cell proliferation was measured with a CCK8 assay following the manufacturer’s protocol (Dojindo Molecular Technologies, Inc., Kumamoto, Japan). The IC50 was calculated using Prism software using the following algorithm:$$\mathrm{Cell}\;\mathrm{Viability}=\frac{100}{1+10^{\left(\mathrm{LogIC}50-\mathrm{Concentration}\right)\times \mathrm{Hillslope}}}$$

### 2.7. Clonogenic Survival Assays

HS683 cells were prepared in a 200 μL centrifuge tube and treated with BPA or TMZB at gradient concentrations in 0.0–1.0 μg/mL for 30 min. Cells were irradiated with epithermal neutrons for approximately 30 min to achieve total neutron fluence at 2.4 ± 0.3 × 10^11^ n/cm^2^. After radiation, the drug-containing medium was removed, and cells were incubated in fresh medium for 12 h. A certain number of cells in a single-cell suspension were seeded into 6-well plates to grow colonies. After 2 weeks of incubation, colonies were fixed with 70% ethanol at room temperature for 10 min, stained with 0.1% crystal violet, and then dried overnight. Colonies with more than 50 cells were scored as clonogenic survivors under a bright field microscope.

### 2.8. In Vivo BNCT Tests

PET/CT imaging with 1.5 uCi/g ^18^F-FDG was performed for all the mice anesthetized with 2% isoflurane at 25 °C before BNCT. Mice were anesthetized with an intraperitoneal injection of tribromoethanol at a dose of 150 mg/kg. The mice were then injected i.v. with 50 mg ^10^B/kg of BPA (5% fructose solution) or the TMZB solution at 50 mg ^10^B/kg or TMZ at 600 mg/kg through the tail vein. Then, 90 min later, mice were fixed in the mold and anesthetized for BNCT. Mice heads were treated with 2 kw of neutron fluence for 2 h each time and 4 times for each mouse with 2 days interval in between, to achieve a total neutron fluence of 16.5 ± 0.3 × 10^11^ n/cm^2^. Boron compounds were injected prior to each of the radiation fractions. PET/CT imaging was performed again 31 days after BNCT. Three-dimensional reconstructions of the PET/CT images were performed using AMIDE (Stanford University, Stanford, CA, USA). ROI, SUV values, and tumor volume were then analyzed.

All experiments were repeated at least three times unless noted otherwise. All statistical analyses in this study were conducted by using GraphPad (GraphPad, San Diego, CA, USA). Data are presented as mean ± SD.

## 3. Results

### 3.1. Boron Delivery Capability in GBM Cells

In order to integrate the effects of TMZ into a boron delivery agent for BNCT, we constructed TMZB with the objective that the synthesis routes do not affect the activation of TMZ. TMZB was then synthesized following the established protocol (Figure 1). The chemical structure of this compound was evaluated using NMR and MS (see Appendix A).

The quantity of boron uptake in cancer cells determines the effect of a boronated compound in BNCT. We measured boron uptake in the GBM cells treated with TMZB or BPA in parallel for comparison. Samples prepared from three GBM cell lines U87MG, U251, and HS683 were exposed to TMZB or BPA at a series of concentrations for 12 h. Boron concentrations per cell were gradually increasing when more drugs were administered (Figure 2). TMZB exposure resulted in significantly more boron uptake than BPA in U87MG and U251 cells (Figure 2a,b), suggesting that TMZB has a higher affinity to GBM cells. TMZB uptake was cell-type-dependent. Among the tested GBM cell lines, U251 cells have the highest uptake when treated with TMZB (Figure 2b). The exposure of 1.91 mM TMZB on U251 cells resulted in a boron uptake of 0.326 μg/10^6^ cells, while 1.99 mM BPA resulted in a boron uptake of 0.126 μg/10^6^ cells. By contrast, HS683 cells showed a low uptake of both TMZB and BPA (Figure 2c). GBM is a highly vascularized tumor disease. Thus, endothelial cells are usually enriched in GBM stroma. We compared boron concentrations after exposure to TMZB in HS683 and HUV-EC, and we found that the differences between these two cell types were not significant (Figure 2d).

### 3.2. Cytotoxic Effects of The Tested Compounds

TMZB was designed to label boron on TMZ without affecting its function as a chemotherapy agent for GBM treatment. To test whether TMZB maintains its cytotoxicity, we performed a CCK-8 assay and a clonogenic cell survival assay to measure cell viabilities after treatments with BPA, TMZ, or TMZB (Figure 3a,b). HS683 cells were selected in the following experiments because they took up similar amount of TMZB and BPA, excluding the confounding factor from the differential cellular uptake as observed in other GBM cell lines. The results demonstrated that TMZ and TMZB have similar cytotoxicity. Both drugs significantly inhibited cell viability in a dose-dependent manner in HS683 cells (Figure 3a,b). The IC50 values of BPA, TMZ, and TMZB on HS683 cells were 1.111 mM for BPA, 0.284 mM for TMZ, and 0.194 mM for TMZB (Figure 3a). In the clonogenic cell survival assay, the IC50 values for BPA, TMZ, and TMZB were 2.312 mM, 0.213 mM, and 0.208 mM, respectively, indicating that the cytotoxicity of TMZ was retained in TMZB.

### 3.3. In Vitro Effects of BNCT with the Tested Compounds

The clonogenic assay quantifies reproductive cell survival in vitro, which is the gold standard for measuring the radiosensitivity of cancer cells [32,33]. We employed this assay to determine the effects of TMZB-based BNCT in comparison with other drugs. HS683 cells were pre-treated with various concentrations of BPA, TMZ, or TMZB. After BNCT, all three compounds significantly reduced the number of survived clones formed from clonogenic cells (*p* < 0.05; Figure 3c). Exposure to 1.0 μg/mL TMZB (0.243 mM) can cause 91.2% ± 6.4% of clonogenic cell death after BNCT at a total neutron fluence of 2.4 ± 0.3 × 10^11^ n/cm^2^. By contrast, BPA at 0.239 mM only caused 63.7% ± 6.3% clonogenic cell death, while TMZ at 0.258 mM eliminated 47.9% ± 6.6% of clonogenic cells. Thus, the TMZB-based BNCT was significantly more effective than BPA-based BNCT in destroying clonogenic GBM cells.

### 3.4. In Vivo Boron Distribution

We further determined the boron concentrations in the organs of tumor-bearing mice that were treated with 50 mg ^10^B/kg of BPA (5% fructose solution) or a TMZB solution at 50 mg ^10^B/kg (Figure 4a). In the tumor tissue of the TMZB-treated group, the boron concentration was significantly higher than that of the BPA-treated group (*p* < 0.001), indicating that TMZB can transport more boron to the tumor site. In addition, the distribution of boron in other tissues suggested that TMZB mainly accumulates in the liver at 2 h after administration, while BPA mainly accumulates in the kidney. The boron concentrations in mice tumors at 2 and 6 h after i.v. BPA administration were 16.93 ± 9.75 and 5.07 ± 1.43 μg B/g, respectively. By contrast, TMZB administration at 2 and 6 h gave rise to boron concentrations of 48.68 ± 5.84 and 9.57 ± 5.51 μg B/g in mice tumors, respectively. The tumor/normal brain (contralateral brain) ratios of boron concentrations at 1 and 6 h after treatments were not significantly different between BPA and TMZB (Figure 4b). Notably, TMZB showed a significantly higher tumor/blood ratio than BPA at 1 h after treatment (*p* = 0.01), suggesting superiority in GBM targeting (Figure 4c).

### 3.5. In Vivo BNCT Results

PET imaging with ^18^F-FDG has been used to evaluate the therapeutic response in GBM [34]. We performed serial whole-body PET imaging of three groups of GBM mouse models treated with BPA, TMZ, or TMZB. The first PET imaging was performed at an early stage of tumor development (day 38; Figure 5c). The second PET imaging was carried out at 31 days after BNCT treatment (Figure 5c). The mice with treated compounds and BNCT did not show obvious health issues during this period. 

We used a reference region to evaluate the ^18^F-FDG uptake according to published methods [35]. SUV data and tumor volume before and after BNCT were measured using AMIDE software (see Appendix A). After BNCT treatment, ^18^F-FDG uptake in BPA- and TMZB-treated tumors was significantly lower than that in the TMZ-treated group (Figure 5e). After quantification, the tumor volume was decreased by 58.2% following TMZB-based BNCT administration, while the decrease was 35.2% for BPA. The SUV max was also significantly reduced for both BPA- and TMZB-based BNCT (Figure 5d). Consistent with clonogenic cell survival results in vitro, TMZB also showed significantly better efficacy in BNCT than BPA in vivo (Figure 5e). In addition, we monitored the body weight of the GBM mouse models, and the treatments did not induce severe health hazards that affected their body weight (Appendix A).

## 4. Discussion

GBM is universally and rapidly fatal despite radical therapies with surgery and chemoradiations [7]. Various treatment strategies have been tested or are under development to overcome the dramatic therapy resistance of GBM [13,14,15,16,17,18,27]. BNCT is a modality of binary radiation therapy and has shown promising effects in GBM treatment [19,20,24,27]. BNCT exploits heavy charged particles from nuclear reactions occurring in tumor cells through the neutron irradiation of boron-enriched tumors, leading to the efficient destruction of radioresistant and disseminated GBM cells while sparing the surrounding normal tissues. Studies have suggested that a single session of BNCT may be equally effective as a conventional fractionated x-ray radiotherapy for GBM [20]. Therefore, there is a lot of interest in further developing BNCT for GBM treatment.

Efficient boron delivery agents are the key to the successful application of BNCT in GBM treatment. Ideally, the candidate agents for BNCT can penetrate the blood–brain barrier and transport sufficient boron to GBM cells selectively. In this study, we constructed and synthesized a boron-enriched derivative of TMZ using phenylboronic acid and temozolomide. TMZ is rapidly absorbed by GBM cells and broken down spontaneously to liberate the highly reactive methyldiazonium cation that preferentially methylates DNA in guanine-rich regions. To retain the activity of TMZ in GBM, TMZB synthesis routes avoided disrupting the splitting process of the diazo group on the left side of TMZ. Thus, the methyltetrazine structure in TMZB can increase the fat-soluble end effectively for better permeability through the blood–brain barrier, decompose diazomethane free radicals under pH conditions in GBM, and methylate DNA for GBM toxicity.

BPA is the drug of choice for the BNCT of GBM in clinical trials, so we investigated TMZB in parallel with BPA. TMZB showed several advantages over BPA in the BNCT treatment of GBM. First, TMZB has a higher uptake level in GBM cells in vitro and in vivo. Compared to BPA, U251 and U87MG cells took up more boron when treated with TMZB. However, the low boron uptake in HS683 suggested that the selection of GBM patients with certain biological features, such as dysregulated MDR1 gene expression [36], for TMZB-based BNCT is necessary. HUV-EC cells have a similar uptake of TMZB to HS683 (Figure 2d), implying that TMZB may inhibit angiogenesis in GBM treatment [37]. To achieve targeted effects in cancer and low toxicity in normal tissues, BNCT prefers high T/B and T/N ratios. In the GBM mouse model, TMZB administration induced higher boron concentrations than BPA, and the T/B and T/N ratios for TMZB are 2.95 ± 1.12 and 3.85 ± 0.83, respectively (Figure 4). By contrast, they are 2.26 ± 1.56 and 2.61 ± 1.98 for BPA. Thus, TMZB has better GBM affinity and quicker blood clearance than BPA. Second, TMZB retains cytotoxicity for GBM treatment. The IC50 of TMZ and TMZB were comparable (Figure 3a,b). In the BNCT treatment of HS683 models, TMZB showed better efficacy than BPA (Figure 3c and Figure 5d). Because HS683 took up similar levels of both compounds (Figure 2c), the enhanced effects from TMZB may have resulted from its intrinsic toxicity, confirming again that TMZB is a dual-function agent in boron delivery and GBM toxicity. Third, TMZB showed better efficacy in the BNCT of GBM models in vitro and in vivo. PET imaging with ^18^F-FDG has been frequently used for the assessment of therapeutic responses in cancer due its advantages in 3D reconstruction [38]. In this study, BNCT effects in GBM mouse models were measured using PET imaging too. At 31 days after neutron irradiation, the tumor volume decreased by 58.2% for TMZB and 35.2% for BPA, suggesting that TMZB may perform better in BNCT treatment for GBM. At last, although TMZB showed better overall effects, it is difficult to separate the chemotherapeutic effect from the BNCT effect.

GBM is a highly heterogeneous and invasive disease. Boron delivery agents such as BPA usually cannot reach all GBM cells at the BNCT-effective concentrations. Notably, the range of heavy charged particles from BNCT is only about 10 micro meter, so the neighbor GBM cells cannot be eliminated by the released particle radiations from the boron-enriched GBM cells during BNCT. This is a general issue in developing boron delivery agents for GBM treatment. TMZB may overcome this issue because it retains toxicity from the first-line chemotherapy drug TMZ, and thus it can kill GBM cells without boron-induced radiation effects in BNCT.

In recent years, more and more boron delivery agents for BNCT have been developed [19,20]. However, for GBM, new types of boron carriers are restricted by the blood–brain barrier, impeding the progress of BNCT in clinical applications. The results of this study demonstrated that TMZB with dual function in GBM toxicity and targeted boron delivery can enhance the efficacy of BNCT, and further development of TMZB in clinical settings may promote the important role of BNCT in GBM treatment.

## 5. Patents

A patent application was filed with the China National Intellectual Property Administration.

## Figures and Tables

**Figure 1 cells-11-01173-f001:**
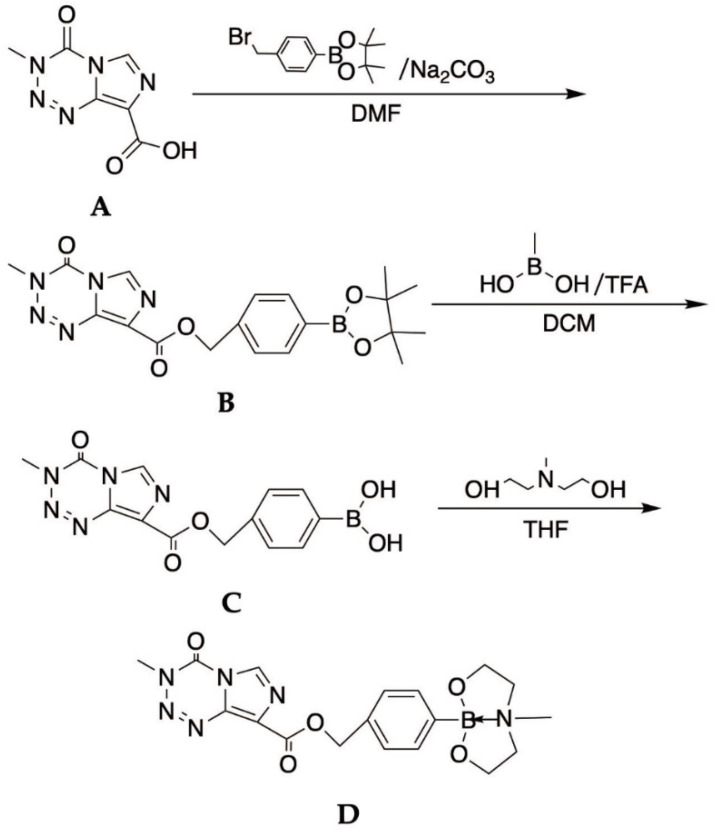
The synthesis routes and chemical structure of TMZB.

**Figure 2 cells-11-01173-f002:**
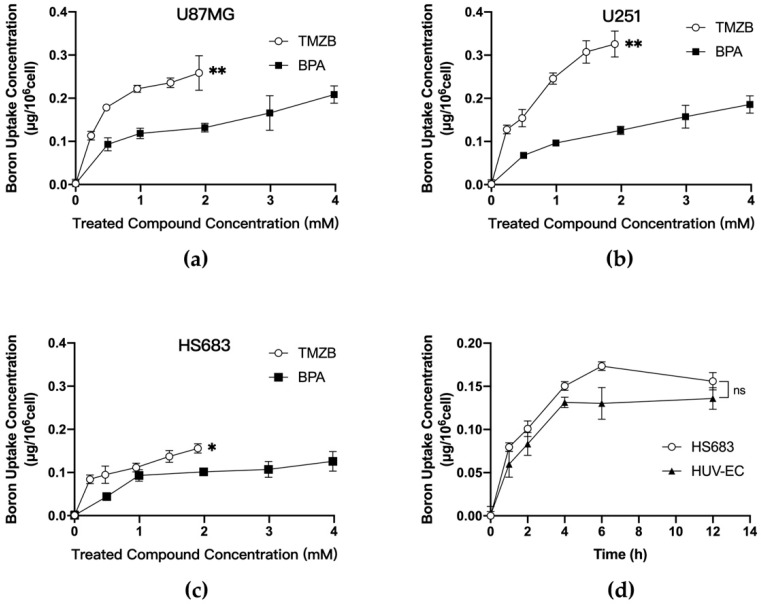
Boron concentrations after exposure of GBM cells to the tested compounds. (**a**) Cellular uptake of TMZB and BPA in U87MG; (**b**) cellular uptake of TMZB and BPA in U251; (**c**) cellular uptake of TMZB and BPA in HS683; (**d**) time-dependent cellular uptake of TMZB (1.91 mM) in HUV-EC and HS683 cells. Boron concentrations were measured using ICP-AES. Bars represent mean ± SE based on 3 biological repeats. The statistical comparisons between curves were carried out by use of the Wilcoxon signed rank test. “ns” stands for not significant. * *p* < 0.05, ** *p* < 0.01.

**Figure 3 cells-11-01173-f003:**
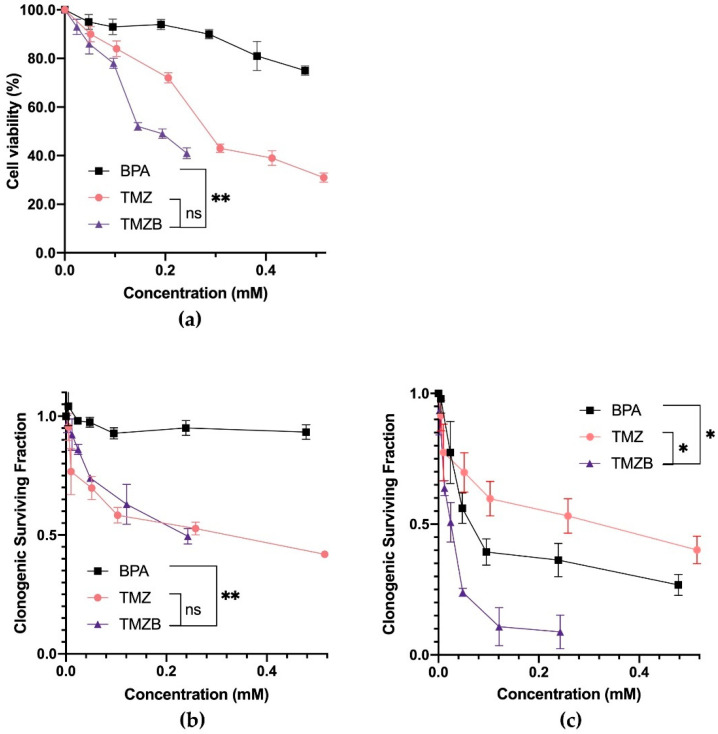
Antitumor activity of BPA, TMZ, and TMZB with or without BNCT. (**a**) HS683 cell viability was measured with a CCK8 assay, where cells were treated with 1.0 μmol/mL of BPA, TMZ, or TMZ-B. (**b**) Clonogenic surviving fractions of HS683 cells treated with various concentrations of BPA, TMZ, or TMZB without BNCT. (**c**) Clonogenic surviving fractions of HS683 cells treated with various concentrations of BPA-, TMZ-, or TMZB-based BNCT. The IC50 was calculated using Prism8 software. The statistical comparisons between curves were carried out using the Wilcoxon signed rank test and survival curve. * *p* < 0.05, ** *p* < 0.01.

**Figure 4 cells-11-01173-f004:**
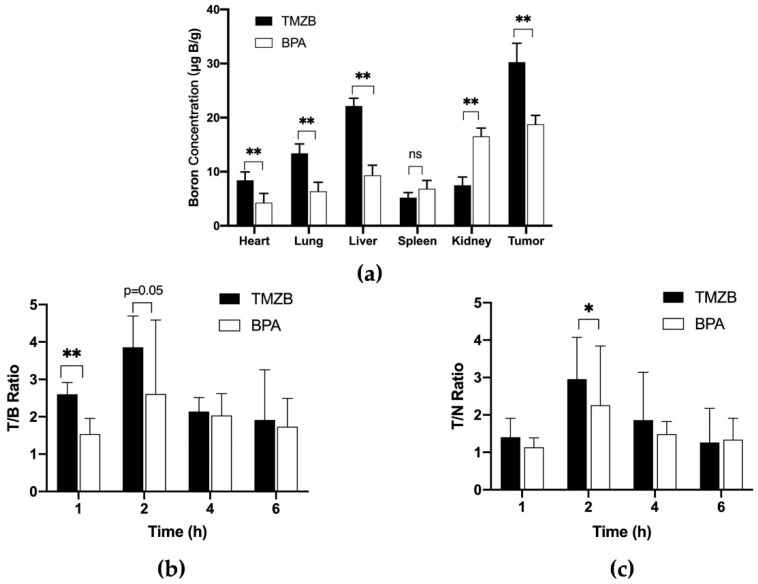
In vivo distribution of boron delivered by TMZB and BPA (50 mg ^10^B/kg). Mice were treated with TMZB or BPA for various lengths of time. Boron concentrations in the heart, lung, liver, spleen, kidney, and tumor were analyzed using ICP-AES. (**a**) Boron concentrations in the heart, lung, liver, spleen, kidney, and tumor after TMZB or BPA injection for 2 h. (**b**) Tumor/blood ratio after TMZB or BPA injection for 1, 2, 4, or 6 h. (**c**) Tumor/normal tissue ratio after TMZB or BPA injection for 1, 2, 4, or 6 h. Mean values with standard error are shown. Statistical comparisons were carried out with a paired *t*-test. * *p* < 0.05, ** *p* < 0.01.

**Figure 5 cells-11-01173-f005:**
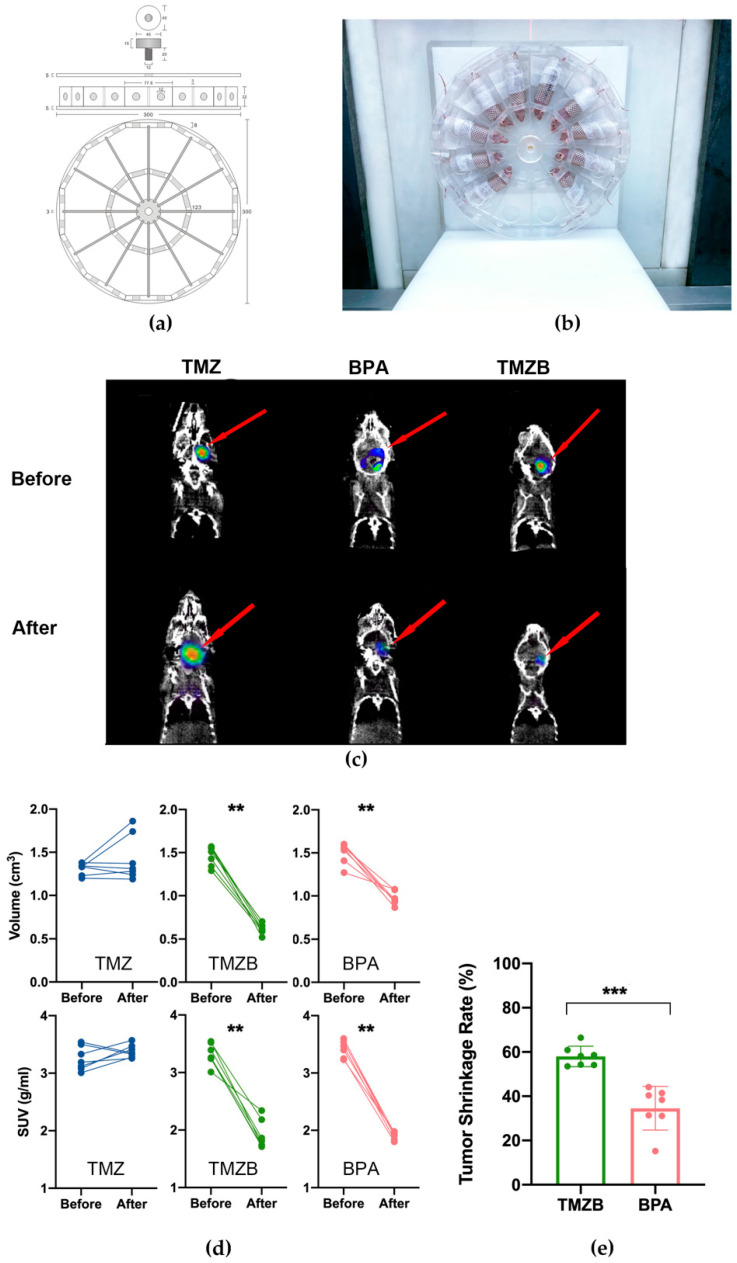
Evaluation of BNCT response of GBM mouse models via PET/CT imaging. (**a**) Schematic illustration for the mice immobilization mold. (**b**) Image of mice in the manufactured device for neutron irradiation in BNCT. (**c**) Representative PET/CT images before and after BNCT treatment (at 31 days after BNCT treatment). (**d**) Volume and SUV change in GBM tumors in each mouse before and after BNCT treatments with BPA (50 mg ^10^B/kg in 5% fructose solution), TMZB solution (50 mg ^10^B/kg), or TMZ at 600 mg/kg. Certain numbers of mice were tested in each treatment group: n_TMZ_ = 7, n_TMZB_ = 8, n_BPA_ = 8. (**e**) Tumor shrinkage rate after BNCT with TMZB or BPA. Statistical comparisons were carried out with a Mann–Whitney test and a two-way ANOVA. ** *p* < 0.01, *** *p* < 0.001.

## Data Availability

The data presented in this study are available on request from the corresponding author.

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
