# Peer review of "A Boronated Derivative of Temozolomide Showing Enhanced Efficacy in Boron Neutron Capture Therapy of Glioblastoma"

_cells, 2022, doi:10.3390/cells11071173_

Round 1
Reviewer 1 Report
This study is original, timely, scientifically sound and contributory to the optimization of BNCT for GBM. In my view, the answer to cancer therapy will be the use of combined therapies. This study proposes a boron carrier (TMZB) that allows for the combination of the therapeutic efficacy of BNCT and of TMZ. I recommend publication after the authors have addressed the following issues:
1) Line 61: capture instead of absorb
2) Line 128: "and the bregma point was selected just above the bregma
point." ?? Re-write
3) Line 164: "Drug-containing medium was removed 12 hours later"...Please explain the choice of this time-point to measure boron concentration in cells. Cells were irradiated at 30 minutes of incubation.
4) Throughout the manuscript I am concerned with how the authors express the dose of BPA and TMZB. For these doses to be comparable they should be expressed as mg 10B/kg not as mg BPA/kg or mg TMZB/kg. Please clarify/correct/discuss.
5) Were the cells irradiated in the presence of the boron compound?
6) Please give some information on the absorbed dose values for the BNCT dose components in each case. The authors only quote total neutron flux (I imagine they mean total neutron fluence).
7) Line 191: Here the authors do quote boron dose: "50 mg 10B/kg of BPA (5% fructose solution) or TMZB solution, or TMZ at 50 mg 10B/kg." However, the dose of TMZ can´t be expressed as mg 10B/kg.
8) Lines 194-195: Was the boron compound injected prior to each of the fractions? Please clarify.
9) Line 220: "Compared to BPA, exposure of equal amount of TMZB (0.8 mg/g)" - equal amount of boron??
10) Fig. 2: What do the authors mean by chemical concentration? Compound concentration or boron concentration? For the data to be comparable the dose should be quoted in mg 10B/kg. In the case of TMZ they should use the equivalent dose of TMZ corresponding to TMZB (TMZB that delivers the same amount of boron as the BPA dose employed). Please clarify/correct/discuss. This issue should be addressed throughout the manuscript.
11) Figure 3b: the left panel (no BNCT? Compound alone?) for BPA indicates a reduction in clonogenic survival due to BPA alone. Is this correct? The reduction for BPA alone is greater than for TMZB alone? And similar to TMZ alone? Time after BNCT? What concentration of the boron compound do panels (b) refer to?
12) Line 304: "while it was 35.2% for BPA". I assume the authors mean while the decrease was 35.2% for BPA.
13) The authors do not address the very important issue of toxicity/radiotoxicity associated to therapeutic efficacy. Please discuss and provide any available information.
14) Line 301: (see Figure.S2) should be replaced by (see Figure S3).
15) Figure S3: Time after BNCT? Please include this information.
Author Response
This study is original, timely, scientifically sound and contributory to the optimization of BNCT for GBM. In my view, the answer to cancer therapy will be the use of combined therapies. This study proposes a boron carrier (TMZB) that allows for the combination of the therapeutic efficacy of BNCT and of TMZ. I recommend publication after the authors have addressed the following issues:
Thank you very much for the endorsement of our research.
1) Line 61: capture instead of absorb
Revised. See line 59
2) Line 128: "and the bregma point was selected just above the bregma
point." ?? Re-write
Thanks for your suggestion. We re-wrote it in line 126-127.
3) Line 164: "Drug-containing medium was removed 12 hours later"...Please explain the choice of this time-point to measure boron concentration in cells. Cells were irradiated at 30 minutes of incubation.
Boron delivery agents with small molecules such as BPA were usually applied in vitro for 2-12 hours[1-2]. TMZB is a derivative of TMZ which is in continuous use for a few weeks during chemotherapy. Therefore, we used 12 hours to ensure sufficient cellular uptake.
- Rodriguez, C.; Carpano, M.; Curotto, P.; Thorp, S.; Casal, M.; Juvenal, G.; Pisarev, M.; Dagrosa, M.A. In vitro studies of DNA damage and repair mechanisms induced by BNCT in a poorly differentiated thyroid carcinoma cell line. Radiation and Environmental Biophysics 2018, 57, 143-152, doi:10.1007/s00411-017-0729-y.
- Fukuo, Y.; Hattori, Y.; Kawabata, S.; Kashiwagi, H.; Kanemitsu, T.; Takeuchi, K.; Futamura, G.; Hiramatsu, R.; Watanabe, T.; Hu, N.; et al. The Therapeutic Effects of Dodecaborate Containing Boronophenylalanine for Boron Neutron Capture Therapy in a Rat Brain Tumor Model. Biology 2020, 9, doi:10.3390/biology9120437.
4) Throughout the manuscript I am concerned with how the authors express the dose of BPA and TMZB. For these doses to be comparable they should be expressed as mg 10B/kg not as mg BPA/kg or mg TMZB/kg. Please clarify/correct/discuss.
Thanks for pointing this out. We have revised the manuscript according to your suggestions.
The molecular weight for BPA and TMZB are 209.01 and 412.21, respectively. Therefore, when cells were treated with equal amount of BPA and TMZB (mg/kg), they took up more boron from BPA than TMZB, supposedly. We calculated molar concentration of BPA and TMZB, and re-plotted the relevant figures. The updated figures showed consistent trend with previous format of data in mg/kg (see Figure. 2,3 in the revised manuscript). Thus, although the cells were treated with “mg /kg”, it didn’t affect the conclusion that TMZB showed better efficacy in treating GBM.
5) Were the cells irradiated in the presence of the boron compound?
Yes, we have clarified it in the manuscript.
6) Please give some information on the absorbed dose values for the BNCT dose components in each case. The authors only quote total neutron flux (I imagine they mean total neutron fluence).
Thanks for your suggestion. We calculated the dose following the equation from a publication[3]. The total radiation dose at the tumor was 6.98Gy with neutron fluence at 16.5±0.3×1011 n/cm2 for BPA (50 mg 10B/kg). The total radiation dose at the tumor for TMZB treated at 50 mg 10B/kg was7.45Gy.
Dtotal =RBEH×DH+RBEN×DN +CBEB-10×DB-10 +RBEγ×Dγ
D stands for dose. RBE stands for Relative Biological Effectiveness, CBE stands for Compound Biologican Effectiveness. H stands for Hydrogen and N stands for Nitrogen. γ stands for γ ray.
RBEH =0.9. RBEN =0.185. CBEB-10=5.48. RBEγ=0.52. DB-10=0.808 Gy, DH = DN =1.36 Gy. Dγ for BPA was 2.07 Gy.
- Bleuel, D.L. Determination and production of an optimal neutron energy spectrum for boron neutron capture therapy. 2003.
7) Line 191: Here the authors do quote boron dose: "50 mg 10B/kg of BPA (5% fructose solution) or TMZB solution, or TMZ at 50 mg 10B/kg." However, the dose of TMZ can´t be expressed as mg 10B/kg.
Corrected accordingly. See line 199-200.
8) Lines 194-195: Was the boron compound injected prior to each of the fractions? Please clarify.
Thanks for pointing this out. We have clarified it in the manuscript. See line 203-204.
9) Line 220: "Compared to BPA, exposure of equal amount of TMZB (0.8 mg/g)" - equal amount of boron??
We agreed that equal boron concentration would be better for comparison, and we have revised the figures and the text accordingly.
10) Fig. 2: What do the authors mean by chemical concentration? Compound concentration or boron concentration? For the data to be comparable the dose should be quoted in mg 10B/kg. In the case of TMZ they should use the equivalent dose of TMZ corresponding to TMZB (TMZB that delivers the same amount of boron as the BPA dose employed). Please clarify/correct/discuss. This issue should be addressed throughout the manuscript.
Sorry for the unclear description. In Figure 2, chemical concentration means “treated compound concentration”, because we treated cells with various compound concentrations and measured cellular uptake at boron concentrations by ICP-AES. We also calculated the molar concentrations of the compounds that treated on cells, which didn’t change the trend of cellular uptake of boron concentrations with these compounds. We have updated the figure labels and re-written the text for better expression.
11) Figure 3b: the left panel (no BNCT? Compound alone?) for BPA indicates a reduction in clonogenic survival due to BPA alone. Is this correct? The reduction for BPA alone is greater than for TMZB alone? And similar to TMZ alone? Time after BNCT? What concentration of the boron compound do panels (b) refer to?
Sorry for the unclear description of the results. We have clarified it in the manuscript. Figure 3a only tested the compound alone effects. Figure 3b,c had samples treated with BNCT. In Figure 3b-c, colonies were fixed 2 weeks after BNCT, cells were treated with 1.0 μg/mL of the tested compounds, so it means 0.243 mM for TMZB, 0.478 mM for BPA, 0.515mM for TMZ.
In the updated Figure 3, we removed the images of Figure 3b, and showed compound alone effects from CCK8 assay and clonogenic assay (Figure 3.a,b). Both results showed that the toxicity of TMZB was comparable to TMZ, but BPA showed little toxicity.
12) Line 304: "while it was 35.2% for BPA". I assume the authors mean while the decrease was 35.2% for BPA.
Yes, we have revised it accordingly. Thanks.
13) The authors do not address the very important issue of toxicity/radiotoxicity associated to therapeutic efficacy. Please discuss and provide any available information.
BPA has little toxicity as a commonly used boron delivery agent based on an amino acid (Figure 3). TMZB showed cytotoxic effects of cancer cells in vitro as expected because this compound is a derivative of TMZ (Figure 3a,b). In the GBM mouse model, 10 mice for each group were treated with 50 mg 10B/kg of TMZB, 50 mg 10B/kg of BPA, and 600 mg/kg TMZ, respectively. Only 3 mice were dead at 31 days after treatment:One from BPA group, two from TMZ group. The rest of mice did not show significant symptoms of drug-induced toxicity with or without BNCT. The body weight of these mice also suggested toxicity of these compounds were low (see following graph, Figure S2).
Figure S2. Body weight changes of TMZB, TMZ, BPA and control group.
14) Line 301: (see Figure.S2) should be replaced by (see Figure S3).
Corrected. See line 315
15) Figure S3: Time after BNCT? Please include this information.
Information included “measured at 31 days after BNCT treatment”. See line 325 and supplemental figures.
Reviewer 2 Report
The manuscript by Xiang et al describes an extensive study on a boronated derivative of the antitumor drug temozolomide. The work shows a large amount of data but requires some improvements regarding the presentation of the results and some clarifications on the general plan of the work.
A major concern I have regards the choice of a derivative of temozolomide as boron agent. Temozolomide is usually quite well tolerated and has the advantage to cross the BBB. However, for BNCT, the administration of a huge amount of boron carrier (e.g. BPA) is required. As the TMZB perform better than BPA but not to a such extent to imagine the administration of a small amount of TMZB, the problem of the toxicity of the drug cannot be underestimated, although the results in vivo seem to show that the treatment is well tolerated by mice. I would suggest adding some comments on this point.
The BNCT protocol used 4 irradiations at a distance of 2 days between each treatment, apparently after a single dose of the boron agent (BPA or TMZB). I believe that after 8 days no boron is retained in tumor, at least for BPA. The only data on biodistribution were collected after 2 hours from compound administration. This point needs to be clarified.
A last comment: the in vitro results show that TMZB is more efficient than BPA in BNCT assay; however, it is difficult to separate the chemotherapic from the BNCT effect making it not obvious to conclude that BNCT based on TMZB is better than BPA BNCT. I would ask to add some comments on this point. I am aware that this point is mentioned on page 11, lines 354-358 but I believe that a more extensive discussion would be required
A final question: why the author did not test also intermediate C (Figure 1) as it would be interesting to know its properties
More specific comments:
Page 2, line 60: the use of the word “labeled” may be misleading, I would suggest using enriched instead
Page 2, line 64: the word “concentrated” has not much sense in the phrase, please either delete or use “a sufficient concentration”
Page 2, line 74: “tumor-to-normal tissue ratio at about 3:1” would be better as “tumor-to-normal tissue ratio of at least 3:1”
Page 3, line 103: it is unclear how the reference 30 is connected to the previous sentence, I looked at the article and seems not to be pertinent to the use of chemicals
Page 3, lines 128-129: I do not understand the meaning of the sentence: … and the bregma point was selected just above the bregma point.
Synthesis: please add the yields for each synthetic step. Moreover please check the English style as it needs to be improved,
Page 5, Figure 1: there are some mistakes in the formulas: hydrogens are missing both on the carboxyl group of a and on the methylboronic acid (over the arrow from B to C; the arrow for dative bond in compound D is misplaced; I would also suggest to increase the size of the atom labels
Page 11, line 350: contract should be contrast
Author Response
The manuscript by Xiang et al describes an extensive study on a boronated derivative of the antitumor drug temozolomide. The work shows a large amount of data but requires some improvements regarding the presentation of the results and some clarifications on the general plan of the work.
A major concern I have regards the choice of a derivative of temozolomide as boron agent. Temozolomide is usually quite well tolerated and has the advantage to cross the BBB. However, for BNCT, the administration of a huge amount of boron carrier (e.g. BPA) is required. As the TMZB perform better than BPA but not to a such extent to imagine the administration of a small amount of TMZB, the problem of the toxicity of the drug cannot be underestimated, although the results in vivo seem to show that the treatment is well tolerated by mice. I would suggest adding some comments on this point.
Thank you very much for your comments and suggestions.
BPA has little toxicity as a commonly used boron delivery agent based on an amino acid (Figure 3). TMZB showed cytotoxic effects of cancer cells in vitro as expected because this compound is a derivative of TMZ (Figure 3a,b). In the GBM mouse model, 10 mice for each group were treated with 50 mg 10B/kg of TMZB, 50 mg 10B/kg of BPA, and 600 mg/kg TMZ, respectively. Only three mice were dead at 31 days after treatment. One from BPA group, two from TMZ group. The rest of mice did not show obvious symptoms of drug-induced toxicity with or without BNCT. The body weight of these mice also suggested toxicity of these compounds were low (Figure S2).
The BNCT protocol used 4 irradiations at a distance of 2 days between each treatment, apparently after a single dose of the boron agent (BPA or TMZB). I believe that after 8 days no boron is retained in tumor, at least for BPA. The only data on biodistribution were collected after 2 hours from compound administration. This point needs to be clarified.
Thanks for pointing this out. We treated mice with boron delivery compounds at 2-5h prior to each irradiation, so we tested biodistribution at 2 hours. It has been clarified in the manuscript. See line 204.
A last comment: the in vitro results show that TMZB is more efficient than BPA in BNCT assay; however, it is difficult to separate the chemotherapic from the BNCT effect making it not obvious to conclude that BNCT based on TMZB is better than BPA BNCT. I would ask to add some comments on this point. I am aware that this point is mentioned on page 11, lines 354-358 but I believe that a more extensive discussion would be required.
Thanks for your suggestions. We agreed that it is hard to separate the chemotherapy effect in TMZB base BNCT, because TMZB was designed to have dual effects at the beginning of this project, where both in vitro and in vivo results confirmed it. We further discussed it in the manuscript. See line 379-381.
A final question: why the author did not test also intermediate C (Figure 1) as it would be interesting to know its properties
Good point. We thought about it before. However, the phenylboronic acid part of intermediate C is not stable in water solution so intermediate C will quickly change its colour in wet air.
More specific comments:
Thanks a lot for these comments. We have revised the manuscript accordingly.
Page 2, line 60: the use of the word “labeled” may be misleading, I would suggest using enriched instead
See line 57-58
Page 2, line 64: the word “concentrated” has not much sense in the phrase, please either delete or use “a sufficient concentration”
See line 62
Page 2, line 74: “tumor-to-normal tissue ratio at about 3:1” would be better as “tumor-to-normal tissue ratio of at least 3:1”
See line 73
Page 3, line 103: it is unclear how the reference 30 is connected to the previous sentence, I looked at the article and seems not to be pertinent to the use of chemicals
Removed.
Page 3, lines 128-129: I do not understand the meaning of the sentence: … and the bregma point was selected just above the bregma point.
See line 126-127
Synthesis: please add the yields for each synthetic step. Moreover please check the English style as it needs to be improved,
Thanks, we have edited it. See section 2.4
Page 5, Figure 1: there are some mistakes in the formulas: hydrogens are missing both on the carboxyl group of a and on the methylboronic acid (over the arrow from B to C; the arrow for dative bond in compound D is misplaced; I would also suggest to increase the size of the atom labels
Edited. See page 6
Page 11, line 350: contract should be contrast
Corrected. See line 280
Reviewer 3 Report
This is a significant and interesting paper that a new boron compound derivative of temozolomide. I have a few questions and comments.
1. Figure 2 shows that TMZB has higher affinity to GBM cells and its uptake is cell type dependent. In addition, authors described that selection of GBM patients with certain biological features for TMZB-based BNCT is necessary. I suggest that authors would mention the possible mechanisms showing the differences depending on cell type at the discussion part.
2. About the legend of Figure 3, I think that (b) and (c) are listed in reverse. Please check the figure numbering and legend.
3. In Figure 3(b), why the clonogenic cells of NC would reduce after BNCT? If NC cells do not take up boron, we think that BNCT has no effect.
4. I understand authors selected HS683 cells for In vivo study. Would BNCT show a higher reduction of tumor in U87 and U251 cells?
Author Response
This is a significant and interesting paper that a new boron compound derivative of temozolomide. I have a few questions and comments.
Thank you very much for your review and endorsement of our research.
- Figure 2 shows that TMZB has higher affinity to GBM cells and its uptake is cell type dependent. In addition, authors described that selection of GBM patients with certain biological features for TMZB-based BNCT is necessary. I suggest that authors would mention the possible mechanisms showing the differences depending on cell type at the discussion part.
Agreed, we further discussed it in the manuscript. See line 364.
- About the legend of Figure 3, I think that (b) and (c) are listed in reverse. Please check the figure numbering and legend.
Thanks for pointing it out. We have updated the figures and revised the legends according to another reviewer’s comments.
- In Figure 3(b), why the clonogenic cells of NC would reduce after BNCT? If NC cells do not take up boron, we think that BNCT has no effect.
We have updated figure 3 with better presentation of the data. Indeed, neutron radiation alone only caused little effects with surviving fraction at 98.1% (Figure 3b).
- I understand authors selected HS683 cells for In vivo study. Would BNCT show a higher reduction of tumor in U87 and U251 cells?
Yes, we agreed. According to cellular uptake data in Figure 2, the other cell lines also had higher boron concentrations, which may lead to increased boron neutron capture events.